# Study protocol for a randomized controlled trial: Effect of an everyday cognition training program on cognitive function, emotional state, frailty and functioning in older adults without cognitive impairment

Susana Sáez-Gutiérrez[1]*, Eduardo José Fernández-Rodríguez[1,2], Celia Sánchez-Gómez[2,3], Alberto García-Martín[4], Fausto José Barbero-Iglesias[1,2], Natalia Sánchez Aguadero[1,2]

1 Department of Nursing and Physiotherapy, Universidad de Salamanca, Salamanca, Spain, 2 Institute of Biomedical Research of Salamanca (IBSAL), Salamanca, Spain, 3 Department of Developmental and Educational Psychology, Universidad de Salamanca, Salamanca, Spain, 4 Department of Labour Law and Social Work, Universidad de Salamanca, Salamanca, Spain

* susanasg@usal.es

## Abstract

### Background

Ageing entails changes in complex cognitive functions that lead to a decrease in autonomy and quality of life. Everyday cognition is the ability to solve cognitively complex problems in the everyday world, enabling instrumental activities of life. Benefits have been found in studies using everyday cognition-based assessment and intervention, as the results predict improvements in everyday performance, not just in specific cognitive functions. A study protocol is presented based on assessment and training in everyday cognition versus traditional cognitive stimulation for the improvement of functionality, emotional state, frailty and cognitive function.

### Methods

A parallel randomised controlled clinical trial with two arms will be conducted. It will be carried out by the University of Salamanca (Spain) in eleven centres and associations for the elderly of the City Council of Salamanca. People aged 60 years or older without cognitive impairment will be recruited. Participants will be randomly distributed into two groups: the experimental group will undergo a training programme in everyday cognition and the control group a programme of traditional cognitive stimulation, completing 25 sessions over 7 months. All participants will be assessed at the beginning and at the end of the intervention, where socio-demographic data and the following scales will be collected: The Medical Outcomes Study (MOS), Questionnaire ARMS-e, Everyday Cognition Test (PECC), Scale Yesavage, Test Montreal Cognitive Assessment (MoCA), The Functional Independence Measure (FIM), Fragility Index and Lawton y Brody Scale.

**Data Availability Statement:** No datasets were generated or analysed during the current study. All relevant data from this study will be made available upon study completion to those investigators who request it through FAIRsharing of research data repositories at https://www.re3data.org/.

**Funding:** This study protocol is part of the Research Project on Active Aging with Preventive Physiotherapy PReGe, which has been funded by the Salamanca City Council. Project code: L9AB. The funders had no role in study design, data collection and analysis, decision to publish, or preparation of the manuscript.

**Competing interests:** The authors have declared that no competing interests exist.

## Discussion

The present study aims to improve conventional clinical practice on cognitive function training by proposing a specific assessment and intervention of everyday cognition based on the importance of actual cognitive functioning during the resolution of complex tasks of daily life, giving priority to the improvement of autonomy.

## Trial registration

ClinicalTrials.gov; ID: NCT05688163. Registered on: January 18, 2023.

## Background

For the first time in history, most people worldwide can live to be over 60 years old [1]. This increase in life expectancy, coupled with a declining birth rate, explains why the population has been ageing significantly in recent years. And although ageing is associated with physical, cognitive, psychological and sensory changes [2], diversity in old age is immense. Many international bodies have advocated an approach focused on promoting health far beyond the absence of disease. Of particular note is the approach promoted by the WHO since the 90's, which defines healthy ageing as the process of optimising opportunities for health, participation and security in order to improve quality of life [3,4].

In older adults, one of the causes of decreased functionality is the loss in terms of cognitive performance [5]. Cognition is a complex and dynamic system of interconnected components that allows us to process, organise and use information in order to be functional in our environment, and for this the different cognitive components or processes must be coordinated [6]. Cognitive ageing is characterised by a decline in complex cognitive functions such as attention, working memory, processing speed, executive control and episodic memory [7]. However, other functions are maintained, such as implicit memory, language and acquired knowledge [8]. People in their everyday functioning need these cognitive functions to solve problems or situations, which is where the term "everyday cognition" comes in. This concept establishes the basis for the present study protocol and is defined as the ability of people to solve cognitively complex problems in the real or everyday world [9], allowing them to carry out instrumental activities of daily life in a functional manner. Another subdomain that is present in the resolution of these activities is the psychological or emotional problems that older adults may face in their daily lives, such as depression [10,11].

Occupational therapy is one of the healthcare disciplines that use neurocognitive approaches to maintain or improve functionality in the elderly population [12,13]. Various approaches exist within this discipline to achieve this objective. The most commonly used approach is traditional cognitive stimulation, which involves group activities targeting specific cognitive functions, such as attention or memory. However, this poses a challenge in terms of transferring and generalising the results to everyday functioning. In response, interventions based on everyday cognition are being implemented, specifically, training in complex everyday problems such as managing finances, medication, or transportation, to improve or maintain autonomy in daily life.

There have been studies [14,15] that have established important findings regarding the use of assessments of everyday cognition versus the use of traditional cognitive performance assessment tools. It is considered that the former may be more effective in preventing the

evolution of functional deficits in neurodegenerative diseases; being predictors of the initial stage of the disease and its development.

With respect to the intervention in everyday cognition, we should point out that it has limited scientific evidence and scope, due to the few projects that have been based on this type of intervention model to date. The use of this method could solve the problems of conventional intervention to generalize the results in the daily performance of older adults, since its effectiveness is shown in improvements of specific cognitive functions, without being able to be transferred to the real life of older adults.

Based on the above, we propose as an intervention a training programme in everyday cognition, as opposed to traditional cognitive stimulation programmes. It would involve carrying out an assessment and an intervention focused on the training of basic cognitive functions during the performance of activities that the older adult carries out on a daily basis with materials that are the same or similar to those of real life; as opposed to other traditional programmes that focus on training isolated cognitive functions, without taking into account whether there is the possibility that this type of intervention will subsequently produce benefits in the daily life of the trained subject. The findings will provide important insights for the development of an intervention model that we believe may be more effective in maintaining cognition and functioning in older adults.

## Aim

In line with the above, the main objective of this study is to test our hypothesis that intervention through a specific training programme in everyday cognition will increase the beneficial effects of traditional cognitive stimulation programmes as an intervention on cognitive function, emotional state, frailty and functionality in older adults without cognitive impairment. The objectives are to compare the effects produced by the application of a specific training programme in everyday cognition versus the isolated intervention on cognition by means of a traditional cognitive stimulation programme in older adults without cognitive impairment.

# Material and methods

## Design and setting of the study

An experimental, prospective, randomised, parallel-controlled, clinical trial will be carried out with two arms of fixed allocation with an experimental group and a control group. It will be conducted for one year at the University of Salamanca (Spain) in eleven municipal centres and associations for the elderly belonging to the Salamanca City Council. Participants belonging to the occupational therapy programmes of the Geriatric Revitalisation Project organised by the University of Salamanca, aged 60 years or older without cognitive impairment at the time of inclusion, will be recruited.

## Sample/Participants

**Participants.**  The study population will be composed of people aged 60 years or older without cognitive impairment who voluntarily agree to participate in the Occupational Therapy programme of the Geriatric Revitalisation project organised by the University of Salamanca in collaboration with the Salamanca City Council (Spain), within the framework of the Research in Active Ageing agreement. Regarding the location, the study will be carried out in eleven municipal centres and associations for the elderly belonging to the Salamanca City Council, Spain. The researchers of the University of Salamanca will invite older adults to participate in the study by explaining the details of the clinical trial and they can be included once

they have given their verbal and written consent and meet the following selection guidelines: (A) Inclusion criteria: Being 60 years of age or older, voluntarily signing the informed consent form authorising participation and completing the initial assessment. (B) Exclusion criteria: Not having knowledge of reading and writing or significant deficits in language comprehension, being institutionalised, to have a clinical diagnosis of a neurocognitive disorder included in the DSM-V at the time of the initial assessment and participating in another cognitive stimulation programme. (C) Withdrawal criteria: Dropping out of the programme.

**Randomisation.** Participants voluntarily enroll in one of the eleven groups that belong to different senior centers to receive one of two types of cognitive intervention. In order to ensure the greatest fairness among the different groups, a maximum of 15 and a minimum of 5 participants per group were established. Following recruitment, randomization will be carried out through simple allocation of each group belonging to each senior group by using the Epidat 4.2 program. The eleven intervention groups will be allocated to either the experimental or control group as detailed in Fig 1 of the SPIRIT 2013 registration, interventions and assessments program.

**Blinding.** The work of sequencing, randomisation, recruitment and allocation of the senior centres to each group (experimental or control) will be completed by research staff not involved in the evaluations or interventions of each group, which will avoid possible biases in the study.

The participants will also be blinded on the hypothesis and expectations of the study. In order to minimise any contamination between groups, the evaluation process will be carried out by external research staff who will carry out the measurements, previously educated and trained; with the aim of avoiding subjective biases in the process, as they will be unaware of the intervention group to which each elderly centre is assigned, so that the clinical trial will have a masking of blind evaluation by third parties. In addition, the researchers responsible for the statistical analysis of the study will be blinded, with the intention of increasing the rigour of the study process, thus increasing the scientific quality.

**Sample size.** The sample size has been estimated using Epidat 4.2. A one-sided approach has been assumed with a confidence level of 95% and a statistical power of 80%. It has been decided to take equal sizes in the two groups (i.e. R = n2/n1) and the Yates correction will not be applied. The sample size calculation was based on the results of a similar RCT, specifically the study conducted by Fernández et al. in 2018 [16] which compared the outcomes of a traditional cognitive stimulation intervention versus an intervention based on everyday cognition. The primary variable chosen in both studies was everyday cognition. The study found that the experimental group (those who received everyday cognition training) demonstrated an improvement of 2.57 points in the primary variable, while the control group (those who received traditional cognitive stimulation) only demonstrated an improvement of 0.39 points. With an expected population loss of 20%, the final sample size was n = 99 participants, with 50 in the experimental group and 49 in the control group.

## Procedures and data collection

**Evaluations and study plan.** Participants will be assessed at two points during the study: the baseline assessment at the beginning of the study and the final assessment after the end of the intervention. The baseline assessment will take place after recruitment and before randomisation and allocation of subjects to the corresponding group. This initial assessment includes the recording of the independent variables and the primary outcome variables to be studied; the intervening variables will be recorded and several objective assessment tests will be used. Then, after randomisation, the intervention for each group will be carried out. Then, another assessment will be carried out, considered the final assessment, where participants will

| | STUDY PERIOD | | | | | | | |
| --- | --- | --- | --- | --- | --- | --- | --- | --- |
| | Enrolment | Allocation | Post-allocation | | | | | Close-out |
| **TIMEPOINT** | *Month 1* | Month 2 | Month 3-4 | Month 5-6 | Month 7 | Month 8 | Month 9 | *Month 10-12* |
| **ENROLMENT:** | | | | | | | | |
| **Eligibility screen** | X | | | | | | | |
| **Informed consent** | X | | | | | | | |
| **Database creation** | X | | | | | | | |
| **Allocation** | | X | | | | | | |
| **INTERVENTIONS:** | | | | | | | | |
| *Intervention A: Experimental group intervention (Everyday Cognition)* | | | ←—————————————→ | | | | | |
| *Intervention B control group intervention (Traditional Cognitive Stimulation)* | | | ←——————————→ | | | | | |
| **ASSESSMENTS:** | | | | | | | | |
| *Baseline intervening variables* | X | | | | | | | |
| *Outcome variables: everyday cognition, emotional state, cognitive function, functionality, frailty, instrumental activities of daily living* | X | | | | | | | X |
| *Dissemination of results* | | | | | | | | X |

**Fig 1. The schedule of enrolment, interventions, and assessments of SPIRIT 2013.**

be asked to complete the same objective tests as in the initial assessment. The results will be notified by means of an individual and personal report to those persons who expressed the wish at the beginning to receive the information corresponding to the results and evolution. All evaluations will be carried out by research staff, previously trained and instructed in this process. This process can be seen in more detail in Fig 2 which includes a flow chart of the study methodology.

**Description of the variables.** The primary variable will be everyday cognition, measured by the Test for the Evaluation of Everyday Cognition (PECC) and cognitive performance, measured by the Montreal Cognitive Assessment Test (MoCA). Secondary variables will be functionality (Functional Independence Measure, FIM), emotional state (Yesavage Scale), frailty (Frailty Index) and independence in instrumental activities of daily living (Lawton and Brody Scale). We will also record the following intervening variables in the clinical history of each participant: social support (Questionnaire MOS) and adherence to treatment (Questionnaire ARMS-e), and other socio-demographic data (age, sex, level of education, marital status and main occupation).

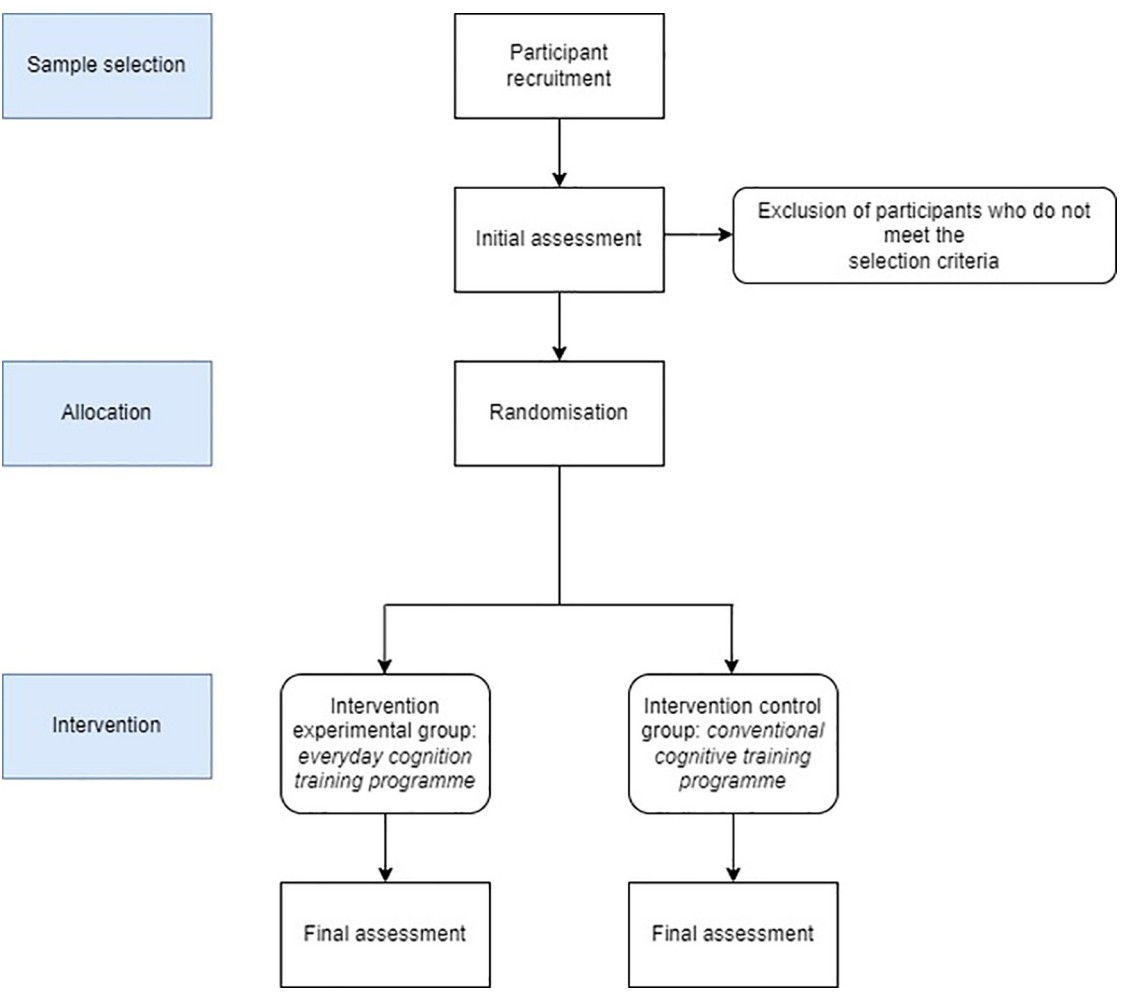

**Fig 2. Study methods flow chart.**

### Tools employed in the evaluation of the variables

1. Test For The Evaluation Of Everyday Cognition (PECC)" [17]: measures the ability of older people to solve 12 real situations grouped into the following areas: medication, administrative management, financial management, meal preparation, transport and shopping, thus allowing to know the functional capacity of their daily life. The administration time is 35 minutes.

2. The Montreal Cognitive Assessment Test (MoCA test) Version 8.3 [18]: allows the detection of mild cognitive impairment (MCI) by assessing executive functions, attention, abstraction, memory, language, visual-constructive abilities, calculation and orientation. The administration time is 10 minutes. The maximum score is 30 points, below 26 points being the cut-off for MCI (in developed countries).

3. Yesavage Geriatric Depression Scale [19]: questionnaire for depression screening in people over 65 years of age. Administration time is 5 minutes. A score above 5 shows moderate depression and above 10 shows severe depression.

4. Functional Independence Measure (FIM)" [20]: measures the level of functionality and assistance given by the caregiver. It assesses activities of daily living subdivided into motor

and cognitive dimensions. Each of the 18 items has a maximum score of 7 and a minimum score of 1, making a total of 126 items.

5. "FRAIL frailty scale" [21]: provides a quantitative measure of frailty in geriatric patients. The maximum score is 5 points.

6. "Lawton and Brody Scale" [22]: designed to assess autonomy in instrumental activities in the elderly population. It takes 4 minutes to administer and is scored 0 or 1 for each of the items, with a total score of 8.

7. "ARMS-e Questionnaire" [23]: analyses in a multidimensional way the lack of adherence in multi-pathological patients, allowing individualised assessment of the barriers detected in adherence problems.

8. "Medical Outcomes Study (MOS) Questionnaire" [24]: allows us to quickly and simply assess social support in a global manner, but also allows us to ascertain the emotional, instrumental, affective and positive social interaction dimensions.

An individual data collection sheet will be used for each participant and recorded in a database designed specifically for this study.

**Interventions.** Two parallel intervention programmes will be designed, coinciding with the two study groups: a training programme in everyday cognition (experimental group) and a traditional cognitive stimulation programme (control group). These programmes will be previously structured and supervised by the team of researchers from the University of Salamanca who form the Geriatric Revitalisation project within the framework of the Research in Active Ageing (PreGe) agreement. The interventions will be carried out on a face-to-face basis in the eleven centres or associations of elderly people in the same city, which voluntarily participate in the Occupational Therapy programme taught by the main researcher of the study. Each centre will have a weekly intervention session for 7 months, completing 25 sessions. The duration of each session will be 50 minutes.

The working technique will be group work, but the activities to be carried out will be individual. The parts of a standard session will be: 10 minutes for the presentation of the session and the explanation of the activity by the researcher, 30 minutes to carry out the activity individually and the last 10 minutes for the correction of the activities, farewell and final feedback. The rooms in the senior citizens' centres where the interventions will be carried out will be similar in terms of infrastructure, being in all cases a suitable environment for the optimal development of the intervention sessions.

As we have pointed out, although the number and duration of the sessions will be the same in both groups, the intervention to be carried out will be different in each study group and the participants will not know to which group they belong:

A. The training programme in everyday cognition will be carried out with the experimental group. For this intervention, materials will be designed and used that are as similar as possible to those that the older adult might encounter in everyday tasks or when facing the resolution of different problems that may arise in their daily lives; with the ultimate goal of bringing the intervention closer to the real life of the participants. In these tasks, participants must make use of cognitive functions such as attention, reasoning, working memory, planning or processing speed, during the development of instrumental activities of daily life such as: meal preparation, housekeeping, use of transport, shopping, use of the telephone, medication, financial management and access to information and current affairs. The proposed activities include context and significance, as they are problems or actions that the person must solve in their daily life in order to maintain their personal autonomy and

independence, in order to be able to live in society. Currently, there is little information in our country about training in everyday cognition. The study carried out by Sánchez et al. in 2019 [25] and the one published by the same authors in 2021 [26], which introduce practical examples of typical sessions, have been taken as a reference for the creation of a training booklet on everyday cognition. We can observe in the first annex to S5, one of these training activities in everyday cognition. In this case, training in medication taking and adherence to treatment will be worked on. In the activity, a practical scenario is explained in which Marisa (fictitious person) has gone to the doctor and has been given a leaflet that she has to read and understand in order to answer the questions correctly. In the second task, participants have to read, understand and memorise a medication leaflet and answer some questions without rereading the information on the leaflet.

B. The traditional cognitive stimulation programme will be carried out with the control group. Tasks focused on specific cognitive functions such as attention, memory, orientation, praxis, calculation, visual perception, executive functions and reasoning will be carried out. For the creation of the cognitive stimulation notebook, reference was made to the study by Calatayud et al. in 2020 [27], who used a "neuronal activation" notebook. The cognitive training notebook includes cards in paper format, where each of the cognitive functions mentioned above are worked on. An example of this is the sample session shown in the second annex of S5, in which attention is worked on by describing what is observed in an image, and then participants are asked to answer questions about it.

You can see an example of a session in the document (S5 File).

## Data analysis

To test the normal distribution of data, a histogram will be used. Variables that have a normal distribution shall be defined by the mean and standard deviation, while variables that do not follow a normal distribution shall be defined by the median and the interquartile range. The effects of the intervention on primary outcomes (everyday cognition and cognitive function) and secondary outcomes (functionality, emotional state, and fragility) will be examined through a two-way analysis of covariance (ANCOVA), with the intervention group (control or experimental) included as a fixed factor. The differences between initial results and those after intervention, as both the dependent variable and covariates, will include intervening study variables such as age, gender, education level, marital status and occupation. The difference in means will be calculated using Cohen's d to evaluate effect size. Group differences in intervening variables will be analyzed using independent samples t-tests for continuous variables (e.g., age). The chi-square test will be used to investigate group differences in categorical variables (e.g. gender). Significant values will be considered for p-values less than 0.05, with a 95% confidence interval. IBM SPSS Statistics software version 28.0.1 will be used for statistical analysis".

## Ethical aspects

The study was approved by the Clinical Research Ethics Committee of the Area of Health of Salamanca (ID: PI 902) (S2 File) (S3 File), having obtained the prior written informed consent of the study subjects and in conformance with the Helsinki Declaration (S4 File). The participants will be informed of the objectives of the project and the risks and benefits of the interventions that will be carried out during the work. Participants will be fully informed, and we will guarantee that their data will be treated with the utmost confidentiality throughout the study and also after its completion. Consequently, data and information will not be shared with third parties. Researchers involved in the study will strictly adhere to professional

confidentiality and the confidentiality of the participants will be guaranteed at all times in accordance with the laws on the protection of personal data and biomedical research presented in the provisions of Organic Law 3/2018 of December 5 on the Protection of Personal Data and the Guarantee of Digital Rights, Regulation (EU) 2016/679 of the European Parliament and Council of 27 April 2016 on data protection (GDPR), and under the conditions established by Law 14/2007 on biomedical research.

Significant modifications to the protocol (such as changes in the tools of evaluation, modifications to the selection criteria or to the interventions) will be communicated immediately to the Ethics Committee.

And since this is a randomised clinical trial, it follows the CONSORT guidelines, and it was registered. TRIAL REGISTRATION: ClinicalTrials.gov; ID: NCT05688163.

### Rigour

This study protocol has been developed following the evidence-based recommendations of the SPIRIT 2013 Statement [28] on the minimum content of a clinical trial protocol (S1 File). And the study design also follows the recommendations of the CONSORT 2010 [29] guidelines for the conduct of parallel-group randomized controlled clinical trials (RCTs).

The TIDieR checklist is added as a tool to describe and replicate the intervention proposal (S6 File).

Data availability: Upon completion of the study, the data collected here will be made available to those investigators who request it through FAIRsharing of research data repositories at https://www.re3data.org/.

### Discussion

The expected results of this study can be transferred to clinical practice, incorporating a specific assessment of everyday cognition, as well as a training programme for everyday cognition. This study aims to improve conventional clinical practice on the training of cognitive functions by proposing a new tool based on the importance of real cognitive functioning or everyday cognition, assessing and intervening on the cognition of older adults during the resolution of complex tasks in their daily lives, giving full priority to the main objective of occupational therapy, which is the gain in terms of autonomy and functionality, and thus improving the quality of life of older adults.

We would like to highlight the importance of promoting the use and creation of assessment tests focused on everyday cognition from occupational therapy, as it gives a global vision of the person's current occupational performance.

Furthermore, as we pointed out, we hope that the results of the study will solve shortcomings found in our daily clinical practice, with interventions based on traditional cognitive stimulation; such as problems in generalising the increase in scores or improvement of cognitive performance in the daily performance of older people. It is important to create specific programmes for everyday cognition that focus on specific training in the different instrumental activities of daily living.

It is important to point out that the short history of research in everyday cognition makes it difficult to carry out an intervention based on scientific evidence. Regarding assessment tests, it should be noted that more and more researchers support the validity of the use of assessments based on everyday cognition for the study of older people with and without cognitive impairment, being a good predictor for the diagnosis and prognosis of neurodegenerative disease [30,31].

### Limitations

The study follows all CONSORT recommendations, however, due to the nature of the intervention the principal investigator is the same person who will carry out the intervention process. However, the participants will be blinded. And, in addition, to minimise any contamination between groups, the researchers responsible for randomisation, assessments and analysis will be blinded.

### Dissemination plan

Dissemination of the results of the study will be carried out with the intention of ensuring maximum visibility. The results will be published in open access scientific journals. There will be an initial publication of the primary results and the publication of secondary results is foreseen. The results will also be presented at national and international conferences and seminars.

### How potential changes in the study will be approached

Significant modifications to the protocol (such as changes to the assessment tools, modifications to the selection criteria or to the interventions) shall be reported immediately to the bioethics committee for approval.

## Conclusions

This manuscript presents the protocol of a study aimed at evaluating the outcomes of an intervention model based on everyday cognition training on cognitive performance, functioning, emotional state and frailty in cognitively unimpaired older adults.

## Supporting information

**S1 File. SPIRIT 2013 checklist: Recommended items to address in a clinical trial protocol and related documents.**
(DOCX)

**S2 File. Acceptation of the ethics committee of the University of Salamanca.**
(PDF)

**S3 File. Research protocol accepted by the ethics committee of the University of Salamanca.**
(PDF)

**S4 File. Informed consent and participant information sheet.**
(PDF)

**S5 File. Sample session: Daily cognitive training programme vs. cognitive stimulation.**
(PDF)

**S6 File. TIDieR checklist.**
(PDF)

## Acknowledgments

The authors would like to thank the University of Salamanca and the Salamanca City Council for their support.

## Author Contributions

**Conceptualization:** Susana Sáez-Gutiérrez.

**Methodology:** Susana Sáez-Gutiérrez, Celia Sánchez-Gómez.

**Project administration:** Eduardo José Fernández-Rodríguez.

**Supervision:** Eduardo José Fernández-Rodríguez, Celia Sánchez-Gómez.

**Writing – original draft:** Susana Sáez-Gutiérrez.

**Writing – review & editing:** Eduardo José Fernández-Rodríguez, Celia Sánchez-Gómez, Alberto García-Martín, Fausto José Barbero-Iglesias, Natalia Sánchez Aguadero.

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
