## [Decision Letter · Decision Letter 0]

16 Oct 2023

PONE-D-23-18288Study protocol for a randomized controlled trial: Effect of an everyday cognition training program on cognitive function, emotional state, frailty and functioning in older adults without cognitive impairmentPLOS ONE

Dear Dr. Sáez,

Thank you for submitting your manuscript to PLOS ONE. After careful consideration, we feel that it has merit but does not fully meet PLOS ONE’s publication criteria as it currently stands. Therefore, we invite you to submit a revised version of the manuscript that addresses the points raised during the review process.

We look forward to receiving your revised manuscript.

Kind regards,

Alessandro Rodolico

Academic Editor

PLOS ONE

Journal Requirements:

"The authors would like to thank the University of Salamanca and the Salamanca City Council for their support."

"YES- The present study protocol is part of the Research Project on Active Ageing with Preventive Physiotherapy PReGe, which has been

active ageing with preventive physiotherapy PReGe, which has been funded by the Salamanca City Council.

Salamanca. Project code: L9AB."

"YES- The present study protocol is part of the Research Project on Active Ageing with Preventive Physiotherapy PReGe, which has been

active ageing with preventive physiotherapy PReGe, which has been funded by the Salamanca City Council.

Salamanca. Project code: L9AB."            

6. We note that the original protocol that you have uploaded as a Supporting Information file contains an institutional logo. As this logo is likely copyrighted, we ask that you please remove it from this file and upload an updated version upon resubmission.

Additional Editor Comments :

Dear authors, after careful evaluation, the decision is to invite a major revision for your manuscript. Below are the summarized core areas that require attention:

Study Design and Randomization: Clarify your randomization procedure, particularly in regard to individual vs. center randomization.

Sample Size and Justification: Your sample size calculation is unclear and seems disconnected from previous work in the field.

Statistical Analysis: Provide a more focused and tailored description of your proposed statistical techniques.

Eligibility Criteria: Refine and clarify the eligibility criteria, specifically around the definition of "cognitive impairment."

Blinding: Detail how you plan to evaluate the effectiveness of blinding.

Clarity and Focus: The introduction requires condensation and a clearer focus on what makes your study unique.

Editorial Changes: Please correct all typographical and grammatical errors, including redundant phrases.

We encourage you to address the individual points raised by each reviewer in a comprehensive and detailed manner.

Reviewers' comments:

Reviewer's Responses to Questions

**Comments to the Author**

1. Does the manuscript provide a valid rationale for the proposed study, with clearly identified and justified research questions?

Reviewer #1: Partly

Reviewer #2: Yes

Reviewer #3: Partly

2. Is the protocol technically sound and planned in a manner that will lead to a meaningful outcome and allow testing the stated hypotheses?

Reviewer #1: Partly

Reviewer #2: Yes

Reviewer #3: Partly

3. Is the methodology feasible and described in sufficient detail to allow the work to be replicable?

Reviewer #1: Yes

Reviewer #2: Yes

Reviewer #3: Yes

4. Have the authors described where all data underlying the findings will be made available when the study is complete?

Reviewer #1: Yes

Reviewer #2: Yes

Reviewer #3: Yes

5. Is the manuscript presented in an intelligible fashion and written in standard English?

Reviewer #1: No

Reviewer #2: Yes

Reviewer #3: Yes

6. Review Comments to the Author

You may also provide optional suggestions and comments to authors that they might find helpful in planning their study.

Reviewer #1: This is a pre-post study of a behavioral intervention. While the study may lead to some results of interest, the protocol is written in a boilerplate style, and I had difficulties determining what specifically the authors are going to do.

--Randomization is not clear: what randomization procedure will be used, and what does randomizing individuals/center mean? Is this a stratified randomization within center: will stratified blocks be used? Unclear.

--Sample size is unclear. A reference is given with respect to a prior study and their results. What was their outcome? What were their interventions? There is no evidence given that this study is comparable to this study. And if it was, why should this study be done? And the treatment effect should be determined as a minimally clinically relevant effect size, not the results of someone else's study. Prior studies are used to estimate measures of variability. It is also not clear what the "main variable" is. I need more specificity to understand what the authors are doing here.

--Statistical analysis techniques are just a laundry list of standard techniques. What particular model will be used for what particular analysis? What variables will be contained in the model? The authors mention logistic regression, but I could not find any binary variables at all: the questionnaire metrics appear to be ordinal. No discussion of how to deal with complex correlated ordinal data is presented. I presume each of the questionnaires has been validates in prior studies, but I can't find any information on that.

--In conclusion, this protocol could have been written for any study: what are the unique characteristics about this study that merits publishable protocol?

Reviewer #2: Thank you for giving me the opportunity to review this manuscript.

I think this study is interesting and scientifically sound.

1) In the eligibility criteria, the authors described "a clinical diagnosis of cognitive impairment at the time of the initial

assessment and participating in another cognitive stimulation programme.". However, I think it is still unclear. Please define the criteria more clearly. For example, what kind of diagnostc criteria of "cognitive impairment" will be used? Will patients with subjective cognitive impairment or social cognitive impairment included? Will patients with cognitive impairment due to mental disorders (bipolar disorders, depression, schizophrenia) included? Will participants who take medication be included?

2) in page5, please delete one word of ""prospective" in the sentence of "An experimental, prospective, randomised, parallel-controlled, prospective clinical trial will be carried out with two arms of fixed allocation with an experimental group and a control group"

3) Please describe how to assess the effectiveness of blinding in this study. If the participants will be informed of the objectives of the project and the risks and benefits of the interventions that will be carried out during the work, it is possible for the participants to assess which groupds they are allocated to after the intervention. Therefore, please descrbe why it is completely possible to blind participants, or how to assess the effectiveness of blinding.

I think it is still necessary to revise the manuscript.

Reviewer #3: The proposed protocol is an interesting study and I wish the authors every success with it. I include some comments below that I think will help improve the study:

Introduction

It is unnecessarily long and can be reduced to

a)healthy ageing and being functionally independent is important, as population ages,

b) part of heathy ageing is cognitive performance,

c) this can be 'trained' with traditional (describe) or everyday (describe) cognition tasks,

d) the study aims...

At the moment it includes lengthy information on aspects that are not necessary (e.g. para 1 and 2) and lacks information on aspects that are (e.g. what each cognition training is/does, how one might be better than the other).

Methods

Maybe provide all the information for the reader to verify the sample size calculation.

Clarification that researchers and participants will be blinded to the hypothesis of the study. Obviously, both researchers and participants will know what group they are in, as in they will know the tasks they are asked to do. What they will not know is what the expectation is, i.e. group everyday (doing X tasks) will be better than group traditional (doing Y tasks).

The analysis needs some considerable revisiting to clarify several matters, as quite a few aspects are unclear:

• Why both Kolmogorov-Smirnov and Shapiro-Wilk for normality of distribution, what do you gain? Why not the more widely acceptable approach of one of the two plus visual inspection of e.g. Q-Q plot or histogram?

• The questionnaires appear to be score-based. If you have any qualitative data from e..g interviews or open comments, will this need soe thematic analysis first before it is then converted to frequencies?

• Validity / reliability have been mentioned here, but with practically no information of what is being examined (which variables, measured how, item / rater reliability?). There is practically no information here other than the statistical tests.

• Once you clarify what is compared (for validity and reliability) you then need to consider the tests again, as to whether simply running a correlation is sufficient or you need more analysis. I would strongly suggest that a simple correlation is insufficient to demonstrate validity / reliability.

• It appears you are treating the questionnaires as continuous data (hence the ANOVA analysis). Although this is not uncommon (especially given the sample size and the wider scores the participants can achieve – e.g. https://www.frontiersin.org/articles/10.3389/feduc.2020.589965/full , https://www.nejm.org/doi/full/10.1056/NEJM198408163110705), it might be useful to provide a rationale for that to avoid apprehension to your analysis (e.g. https://online.ucpress.edu/abt/article-abstract/82/5/289/110285/When-ANOVA-Isn-t-Ideal-Analyzing-Ordinal-Data-from?redirectedFrom=fulltext).

• For clarity, I would describe the analysis approach first of what you are trying to do (e.g. compare both groups and pre-post) and then the two statistical approach options (non/parametric).

• Have you considered an ANCOVA for the design you have (e.g. https://www.ncbi.nlm.nih.gov/pmc/articles/PMC6290914/)?

• Correlation and regression have been mentioned here, but with practically no information (which variables are correlated, why). The design is a 2 (group) x 2 (time points), thus looking for differences; what purpose are the correlation and regression serve?

Dissemination

Have you considered non-scientific publications for a practitioner-oriented language dissemination of the study?

Please correct typos:

P7, Line 241 – replace ‘who request at the beginning the wish to receive with either ‘who requested at the beginning to receive’ or ‘ who expressed the wish at the beginning to receive’

P11, Line 432 – replace ‘approch’ with ‘approach

7. PLOS authors have the option to publish the peer review history of their article (what does this mean?). If published, this will include your full peer review and any attached files.

Reviewer #1: No

Reviewer #2: No

Reviewer #3: No

---

## [Author Response · Author response to Decision Letter 0]

8 Nov 2023

COMMENTS FROM REVIEWER 1 Authors’ Responses 

Randomization is not clear: what randomization procedure will be used, and what does randomizing individuals/center mean? Is this a stratified randomization within center: will stratified blocks be used? Unclear. Thank you for your comment. It is true that the initial manuscript does not clearly explain the method for randomization. Each participant will enroll in one of the eleven senior centers where the activity is taking place. To achieve balance across centers, a maximum of 15 participants per group and a minimum of 5 will be set. At this stage, the randomisation of the different activity groups is carried out through simple allocation using the Epidat 4.2 programme. Therefore, it is not stratified randomisation or block randomisation, but rather simple allocation randomisation of the intervention groups.

The revised version adds the following:

“Participants voluntarily enroll in one of the eleven groups that belong to different senior centers to receive one of two types of cognitive intervention. In order to ensure the greatest fairness among the different groups, a maximum of 15 and a minimum of 5 participants per group were established. In order to ensure the greatest fairness among the different groups, a maximum of 15 and a minimum of 5 participants per group were established. Following recruitment, randomization will be carried out through simple allocation of each group belonging to each senior group by using the Epidat 4.2 program. The eleven intervention groups will be allocated to either the experimental or control group as detailed in Figure 1 of the SPIRIT 2013 registration, interventions and assessments program”.

Sample size is unclear. A reference is given with respect to a prior study and their results. What was their outcome? What were their interventions? There is no evidence given that this study is comparable to this study. And if it was, why should this study be done? And the treatment effect should be determined as a minimally clinically relevant effect size, not the results of someone else's study. Prior studies are used to estimate measures of variability. It is also not clear what the "main variable" is. I need more specificity to understand what the authors are doing here. Thank you for your comment.

The sample size calculation was based on reported results from another RCT that measured the same outcomes and used similar interventions. 

Please find the study link attached:

https://scielo.isciii.es/pdf/geroko/v29n2/1134-928X-geroko-29-02-00065.pdf

The interventions analyzed in this study consisted of a control group receiving traditional cognitive stimulation compared to an experimental group receiving traditional everyday cognition. The main outcome found statistically significant differences in the primary variable (everyday cognition). We believe that our study protocol is significant as it provides results on the various interventions not only in terms of cognitive function and daily cognition, but also analyzes how emotional state, frailty, and functionality change depending on the intervention received. Increasing the individual's sense of functionality reflects benefits in other areas, making our findings highly relevant.

To clarify this procedure, the paragraph has been rewritten by introducing answers to the questions identified:

“The sample size has been estimated using Epidat 4.2. A one-sided approach has been assumed with a confidence level of 95% and a statistical power of 80%. It has been decided to take equal sizes in the two groups (i.e. R=n2/n1) and the Yates correction will not be applied. The sample size calculation was based on the results of a similar RCT, specifically the study conducted by Fernández et al. in 2018 (16) which compared the outcomes of a traditional cognitive stimulation intervention versus an intervention based on everyday cognition. The primary variable chosen in both studies was everyday cognition. The study found that the experimental group (those who received everyday cognition training) demonstrated an improvement of 2.57 points in the primary variable, while the control group (those who received traditional cognitive stimulation) only demonstrated an improvement of 0.39 points. With an expected population loss of 20%, the final sample size was n=99 participants, with 50 in the experimental group and 49 in the control group”.

Statistical analysis techniques are just a laundry list of standard techniques. What particular model will be used for what particular analysis? What variables will be contained in the model? The authors mention logistic regression, but I could not find any binary variables at all: the questionnaire metrics appear to be ordinal. No discussion of how to deal with complex correlated ordinal data is presented. I presume each of the questionnaires has been validates in prior studies, but I can't find any information on that. Thank you for your comment because, as you rightly point out, the statistical analysis section was one of the weak points of the study, so we have decided to modify it completely. It has therefore been replaced by the following paragraph:

“To test the normal distribution of data, a histogram will be used. Sociodemographic variables and reference data, such as therapeutic compliance or level of social support, which have a normal distribution, will be defined by the mean and standard deviation, while variables which do not follow a normal distribution will be defined by the median and interquartile range. Qualitative variables in the study will be defined by frequencies and percentages. A thematic analysis will be conducted to identify key patterns and themes within the qualitative data, such as the variable "occupation", collected through an interview. The effects of the intervention on primary outcomes (everyday cognition and cognitive function) and secondary outcomes (functionality, emotional state, and fragility) will be examined through a two-way analysis of covariance (ANCOVA), with the intervention group (control or experimental) included as a fixed factor. The differences between initial results and those after intervention, as both the dependent variable and covariates, will include intervening study variables such as age, gender, education level, marital status and occupation. The difference in means will be calculated using Cohen's d to evaluate effect size. Group differences in intervening variables will be analyzed using independent samples t-tests for continuous variables (e.g., age). The chi-square test will be used to investigate group differences in categorical variables (e.g. gender). Significant values will be considered for p-values less than 0.05, with a 95% confidence interval. IBM SPSS Statistics software version 28.0.1 will be used for statistical analysis”.

Regarding the validation of the questionnaires used, the link to the studies is attached below:

https://pubmed.ncbi.nlm.nih.gov/23110491/

https://www.elsevier.es/es-revista-neurologia-295-articulo-validacion-del-instrumento-montreal-cognitive-S0213485317301020

https://scielo.isciii.es/pdf/medif/v12n10/original2.pdf

https://scielo.isciii.es/scielo.php?script=sci_arttext&pid=S0213-91112009000100010

https://digibuo.uniovi.es/dspace/handle/10651/28782

https://www.elsevier.es/es-revista-atencion-primaria-27-articulo-diseno-validacion-escala-valorar-fragilidad-S0212656717303797

https://www.ncbi.nlm.nih.gov/pmc/articles/PMC6876023/

https://docplayer.es/7229627-Validacion-del-cuestionario-mos-de-apoyo-social-en-atencion-primaria.html

In conclusion, this protocol could have been written for any study: what are the unique characteristics about this study that merits publishable protocol? This study protocol is a pioneer in active ageing research. Not only does it work with an innovative cognitive intervention model, such as everyday cognition, which would provide a solution to the problem that professionals encounter in clinical practice of being able to transfer the benefits reported from traditional cognitive stimulation, currently the most widely used approach, to everyday functioning. Rather, this study aims to go further, offering holistic results of its benefits in the elderly, such as the benefits that this type of intervention can offer in emotional well-being or the reduction of frailty. In addition to the reported benefits in cognitive function and daily functioning, factors that directly impact the quality of life of older people.

COMMENTS FROM REVIEWER 2 Authors’ Responses 

In the eligibility criteria, the authors described "a clinical diagnosis of cognitive impairment at the time of the initial

assessment and participating in another cognitive stimulation programme.". However, I think it is still unclear. Please define the criteria more clearly. For example, what kind of diagnostic criteria of "cognitive impairment" will be used? Will patients with subjective cognitive impairment or social cognitive impairment included? Will patients with cognitive impairment due to mental disorders (bipolar disorders, depression, schizophrenia) included? Will participants who take medication be included? Thank you for your comment. It is true that the concept presented is broad and needs to be narrowed down. Therefore, I propose:

To be replaced “presenting a clinical diagnosis of cognitive impairment at the time of the initial assessment” by "to have a clinical diagnosis of a neurocognitive disorder included in the DSM-V at the time of the initial assessment."

This way, we would include patients with subjective cognitive impairment, social cognitive impairment, or cognitive impairment due to other mental disorders.

Regarding medication use, though it would be interesting to analyse it as a variable in another study, medication intake is not exclusionary in this protocol due to the high number of medications prescribed to the older population.

In page 5, please delete one word of ""prospective" in the sentence of "An experimental, prospective, randomised, parallel-controlled, prospective clinical trial will be carried out with two arms of fixed allocation with an experimental group and a control group" Thank you for your comment. The change has been recorded in the revised manuscript.

Please describe how to assess the effectiveness of blinding in this study. If the participants will be informed of the objectives of the project and the risks and benefits of the interventions that will be carried out during the work, it is possible for the participants to assess which groupds they are allocated to after the intervention. Therefore, please descrbe why it is completely possible to blind participants, or how to assess the effectiveness of blinding. Thank you for your comment. It is important to note that complete blinding of the participants is impossible, as they are provided with an information sheet and informed consent at the beginning of the study, in addition to receiving one of the two types of intervention. Thus, blinding was not feasible in this study. Throughout the study, participants are unaware of the research hypothesis and expectations, such as which of the two interventions is more effective. 

For clarification, the following sentence has been removed from the blinding: “and will be unaware of which group they have been assigned to and thus the intervention they will receive”. The following sentence has been included for clarification: “on the hypothesis and expectations of the study”.

COMMENTS FROM REVIEWER 3 Authors’ Responses 

The proposed protocol is an interesting study and I wish the authors every success with it. I include some comments below that I think will help improve the study:

Introduction

It is unnecessarily long and can be reduced to

a)healthy ageing and being functionally independent is important, as population ages,

b) part of heathy ageing is cognitive performance,

c) this can be 'trained' with traditional (describe) or everyday (describe) cognition tasks,

d) the study aims...

At the moment it includes lengthy information on aspects that are not necessary (e.g. para 1 and 2) and lacks information on aspects that are (e.g. what each cognition training is/does, how one might be better than the other).

 Thank you very much for your comment. It is true that the initial introduction is long and needs to be thoroughly revised.

- Therefore, the following sentence has been removed in paragraph 1: “To this end, it is important that health professionals working with this population start from a perspective far removed from prejudices that assume that increasing chronological age is synonymous with loss of functionality”. 

- The second paragraph has been deleted in its entirety:

“Occupational Therapy, as a health science, aims to improve or enable the participation of people through the therapeutic use of the occupations of daily living (5). This discipline conceives of the older person as an active and participatory member who carries out occupations in the home and community, and not "occupying" as an act of entertainment, but as a guarantee of health preservation”.

- A new paragraph has been inserted: 

“Occupational therapy is a crucial healthcare science in providing functional maintenance to elderly individuals facing cognitive decline. This is achieved through the therapeutic use of daily life activities, which aim to encourage greater participation among the elderly both at home and within the wider community (12,13)”.

- The following has been included to explain what types of intervention will be worked on in the study:

“Various approaches exist within this discipline to achieve this objective. The most commonly used approach is traditional cognitive stimulation, which involves group activities targeting specific cognitive functions, such as attention or memory. However, this poses a challenge in terms of transferring and generalising the results to everyday functioning. In response, interventions based on everyday cognition are being implemented, specifically, training in complex everyday problems such as managing finances, medication, or transportation, to improve or maintain autonomy in daily life”.

- S6 File. TIDier Checklist has been included as supplementary material to verify and clarify the quality of the information offered on the description of the interventions.

Maybe provide all the information for the reader to verify the sample size calculation. Thank you for your comment, to facilitate this information, the sample size paragraph has been rewritten with some clarifying information for the reader: “The sample size has been estimated using Epidat 4.2. A one-sided approach has been assumed with a confidence level of 95% and a statistical power of 80%. It has been decided to take equal sizes in the two groups (i.e. R=n2/n1) and the Yates correction will not be applied. The sample size calculation was based on the results of a similar RCT, specifically the study conducted by Fernández et al. in 2018 (16) which compared the outcomes of a traditional cognitive stimulation intervention versus an intervention based on everyday cognition. The primary variable chosen in both studies was everyday cognition. The study found that the experimental group (those who received everyday cognition training) demonstrated an improvement of 2.57 points in the primary variable, while the control group (those who received traditional cognitive stimulation) only demonstrated an improvement of 0.39 points. With an expected population loss of 20%, the final sample size was n=99 participants, with 50 in the experimental group and 49 in the control group”.

Clarification that researchers and participants will be blinded to the hypothesis of the study. Obviously, both researchers and participants will know what group they are in, as in they will know the tasks they are asked to do. What they will not know is what the expectation is, i.e. group everyday (doing X tasks) will be better than group traditional (doing Y tasks). Thank you for your comment. For clarification, the following sentence has been removed from the blinding: “and will be unaware of which group they have been assigned to and thus the intervention they will receive”. The following sentence has been included for clarification: “on the hypothesis and expectations of the study”.

Why both Kolmogorov-Smirnov and Shapiro-Wilk for normality of distribution, what do you gain? Why not the more widely acceptable approach of one of the two plus visual inspection of e.g. Q-Q plot or histogram? Thank you for your comment, it is true that this is not the best way to defend the normal distribution of data. The sentence has therefore been replaced by “For the descriptive analysis of the data, normality will be checked with the Kolmogorov-Smirnov and Shapiro-Wilk test (n<30). in the revised manuscript by the phrase “To test the normal distribution of data, a histogram will be used”

The questionnaires appear to be score-based. If you have any qualitative data from e..g interviews or open comments, will this need soe thematic analysis first before it is then converted to frequencies? Thank you for your appreciation. 

It is true that we should include the previous thematic analysis in the qualitative data of the study.

The revised version adds the following: 

“ A thematic analysis will be conducted to identify key patterns and themes within the qualitative data, such as the variable "occupation", collected through an interview.” 

Validity / reliability have been mentioned here, but with practically no information of what is being examined (which variables, measured how, item / rater reliability?). There is practically no information here other than the statistical tests. 

Once you clarify what is compared (for validity and reliability) you then need to consider the tests again, as to whether simply running a correlation is sufficient or you need more analysis. I would strongly suggest that a simple correlation is insufficient to demonstrate validity / reliability.

It appears you are treating the questionnaires as continuous data (hence the ANOVA analysis). Although this is not uncommon (especially given the sample size and the wider scores the participants can achieve – e.g. https://www.frontiersin.org/articles/10.3389/feduc.2020.589965/full , https://www.nejm.org/doi/full/10.1056/NEJM198408163110705), it might be useful to provide a rationale for that to avoid apprehension to your analysis (e.g. https://online.ucpress.edu/abt/article-abstract/82/5/289/110285/When-ANOVA-Isn-t-Ideal-Analyzing-Ordinal-Data-from?redirectedFrom=fulltext).

For clarity, I would describe the analysis approach first of what you are trying to do (e.g. compare both groups and pre-post) and then the two statistical approach options (non/parametric) We are very grateful for all your comments, which have allowed us to improve and analyse the statistical analysis section in depth. This has been redone and replaced by the following paragraph: “To test the normal distribution of data, a histogram will be used. Sociodemographic variables and reference data, such as therapeutic compliance or level of social support, which have a normal distribution, will be defined by the mean and standard deviation, while variables which do not follow a normal distribution will be defined by the median and interquartile range. Qualitative variables in the study will be defined by frequencies and percentages. A thematic analysis will be conducted to identify key patterns and themes within the qualitative data, such as the variable "occupation", collected through an interview. The effects of the intervention on primary outcomes (everyday cognition and cognitive function) and secondary outcomes (functionality, emotional state and fragility) will be examined through a two-way analysis of covariance (ANCOVA), with the intervention group (control or experimental) included as a fixed factor. The differences between initial results and those after intervention, as both the dependent variable and covariates, will include intervening study variables such as age, gender, education level, marital status and occupation. The difference in means will be calculated using Cohen's d to evaluate effect size. Group differences in intervening variables will be analyzed using independent samples t-tests for continuous variables (e.g., age). The chi-square test will be used to investigate group differences in categorical variables (e.g. gender). Significant values will be considered for p-values less than 0.05, with a 95% confidence interval. IBM SPSS Statistics software version 28.0.1 will be used for statistical analysis”.

Have you considered an ANCOVA for the design you have (e.g. https://www.ncbi.nlm.nih.gov/pmc/articles/PMC6290914/)?

• Correlation and regression have been mentioned here, but with practically no information (which variables are correlated, why). The design is a 2 (group) x 2 (time points), thus looking for differences; what purpose are the correlation and regression serve?

 Thank you very much for your comment. The ANCOVA model had not been considered at the beginning, but after a thorough review we believe that it is the most accurate model. With the paragraph attached below, we believe that the comments proposed in this section would be resolved.

The revised version adds the following: 

“The effects of the intervention on primary outcomes (everyday cognition and cognitive function) and secondary outcomes (functionality, emotional state and fragility) will be examined through a two-way analysis of covariance (ANCOVA), with the intervention group (control or experimental) included as a fixed factor. The differences between initial results and those after intervention, as both the dependent variable and covariates, will include intervening study variables such as age, gender, education level, marital status and occupation)”.

Have you considered non-scientific publications for a practitioner-oriented language dissemination of the study? Thank you very much for your question. Non-scientific publications of our study among various professionals working on the autonomy of older people from the prevention of cognitive decline, such as occupational therapists, is within our dissemination plans. Among other things, we want to publish the intervention booklet based on everyday cognition that we are carrying out for the experimental intervention group, as a pioneer in the field.

Please correct typos:

P7, Line 241 – replace ‘who request at the beginning the wish to receive with either ‘who requested at the beginning to receive’ or ‘ who expressed the wish at the beginning to receive’

P11, Line 432 – replace ‘approch’ with ‘approach Thank you very much for your corrections, the typos mentioned in the revised manuscript have been corrected.

---

## [Decision Letter · Decision Letter 1]

18 Jan 2024

PONE-D-23-18288R1Study protocol for a randomized controlled trial: Effect of an everyday cognition training program on cognitive function, emotional state, frailty and functioning in older adults without cognitive impairmentPLOS ONE

Dear Dr. Sáez,

Thank you for submitting your manuscript to PLOS ONE. After careful consideration, we feel that it has merit but does not fully meet PLOS ONE’s publication criteria as it currently stands. Therefore, we invite you to submit a revised version of the manuscript that addresses the points raised during the review process.

**The manuscript has been further evaluated by three reviewers, and their comments are available below. Can you please attend to the ongoing concerns raised by Reviewer #3?**

We look forward to receiving your revised manuscript.

Kind regards,

Avanti Dey, PhD

Staff Editor

PLOS ONE

Reviewers' comments:

Reviewer's Responses to Questions

**Comments to the Author**

1. Does the manuscript provide a valid rationale for the proposed study, with clearly identified and justified research questions?

Reviewer #1: Yes

Reviewer #2: Yes

Reviewer #3: Partly

2. Is the protocol technically sound and planned in a manner that will lead to a meaningful outcome and allow testing the stated hypotheses?

Reviewer #1: Yes

Reviewer #2: Yes

Reviewer #3: Partly

3. Is the methodology feasible and described in sufficient detail to allow the work to be replicable?

Reviewer #1: Yes

Reviewer #2: Yes

Reviewer #3: Yes

4. Have the authors described where all data underlying the findings will be made available when the study is complete?

Reviewer #1: Yes

Reviewer #2: Yes

Reviewer #3: Yes

5. Is the manuscript presented in an intelligible fashion and written in standard English?

Reviewer #1: Yes

Reviewer #2: Yes

Reviewer #3: Yes

6. Review Comments to the Author

You may also provide optional suggestions and comments to authors that they might find helpful in planning their study.

Reviewer #1: The authors have responded adequately to all my comments. 

Reviewer #2: This manuscript is a study protocol for a randomized controlled trial: Effect of an everyday cognition training program on cognitive function, emotional state, frailty and functioning in older adults without cognitive impairment. I think the authors fully answered my questions. I think this manuscript would be suitable for publication in this journal.

Reviewer #3: Thank you for amending the manuscript, which I think it is better. There are still some matters that need clarification.

Introduction:

It is still rather long and unclear. The information about the aim of the Occupational Therapy etc is not needed. The issue here is that not only the novelty of the study is lost (which is not relevant to this Journal's publication criteria) but the methods 'justification' in what each measure is doing, is lost.

Methods:

Remove the duplicate 'In order to ensure...' were established'.

Given this is a protocol paper, I think the methods should be crystal clear. Please clarify the below:

'..To calculate the sample size, everyday cognition was selected as the main study variable..' - is that the PECC?

'..improved 2.57 points on the main variable, while the control group improved 0.39 points' - is that points from a questionnaire, e.g. PECC? An expected effect size would be more useful to replicate the calculations -a rough estimation dictates you are expecting a moderate effect, is that correct?

There seems to be some confusion on the 'qualitative' data and I am at least partially to blame for this. My understanding from the original submission was that there was some interview/focus groups qualitative data, hence my suggestion for thematic analysis. It appears that none of the primary or secondary outcomes includes an interview, so there are only quantitative data with possibly different levels of data as their outcome. Therefore, the thematic analysis comment as well as any reference to qualitative data should be removed.

Further, the analysis appears to 'mix and match', while also making assumptions about the data - how do you know, for example, that some data will be normally distributed and others not? Even more so, the variables expected to be normally distributed appear to also be ordinal in nature. I think this needs further revisiting, in order to:

a) consider the variables you have, why they were collected in the first instance and what they will tell you

b) identify what level of data you have for those variables

c) how you will analyse them and why; include any assumptions made or 'if x happens, we will do y'

d) consider the tests to achieve that

At the moment, there seems to be a somewhat random mix of approaches and analyses, with little justification.

These are the two key aspects for e that require clarification and re-working, so it makes the process easier later, as you'll just need to follow this protocol.

7. PLOS authors have the option to publish the peer review history of their article (what does this mean?). If published, this will include your full peer review and any attached files.

Reviewer #1: No

Reviewer #2: No

Reviewer #3: No

---

## [Author Response · Author response to Decision Letter 1]

31 Jan 2024

Dear reviewer 3.

Firstly, I would like to express my gratitude for your valuable revisions. We hope that you will find them sufficient for publication. Please do not hesitate to contact us if you have any questions or if you require further modifications. 

We believe that the manuscript has improved in quality thanks to your input, which will facilitate its continuation once it is put into action.

We will now attach all the responses and modifications we have made.

Sincerily,

Susana Sáez Gutiérrez.

AUTHORS’ RESPONSE REVIEWER 3

COMMENTS FROM REVIEWER 3

Introduction:

It is still rather long and unclear. The information about the aim of the Occupational Therapy etc is not needed. The issue here is that not only the novelty of the study is lost (which is not relevant to this Journal's publication criteria) but the methods 'justification' in what each measure is doing, is lost.

Authors’ Responses

Thank you for your comment. We agree with your evaluation of the changes in the introduction.

Therefore, we have thoroughly revised it to reduce its length and provide greater significance to the topic.

The following sentence has been removed:

“and factors such as the presence of enabling environments and support can help older people to maintain their autonomy”.

The following sentence has been replaced: “Occupational therapy is a crucial healthcare science in providing functional maintenance to elderly individuals facing cognitive decline. This is achieved through the therapeutic use of daily life activities, which aim to encourage greater participation among the elderly both at home and within the wider community (12,13). Various approaches exist within this discipline to achieve this objective”.

For the following:

“Occupational therapy is one of the healthcare disciplines that use neurocognitive approaches to maintain or improve functionality in the elderly population” (12,13).

COMMENTS FROM REVIEWER 3:

Methods:

1. Remove the duplicate 'In order to ensure...' were established'.

2. Given this is a protocol paper, I think the methods should be crystal clear. Please clarify the below:

'..To calculate the sample size, everyday cognition was selected as the main study variable..' - is that the PECC?

'..improved 2.57 points on the main variable, while the control group improved 0.39 points' - is that points from a questionnaire, e.g. PECC? An expected effect size would be more useful to replicate the calculations -a rough estimation dictates you are expecting a moderate effect, is that correct?

There seems to be some confusion on the 'qualitative' data and I am at least partially to blame for this. My understanding from the original submission was that there was some interview/focus groups qualitative data, hence my suggestion for thematic analysis. It appears that none of the primary or secondary outcomes includes an interview, so there are only quantitative data with possibly different levels of data as their outcome. Therefore, the thematic analysis comment as well as any reference to qualitative data should be removed.

Further, the analysis appears to 'mix and match', while also making assumptions about the data - how do you know, for example, that some data will be normally distributed and others not? Even more so, the variables expected to be normally distributed appear to also be ordinal in nature. I think this needs further revisiting, in order to:

a) consider the variables you have, why they were collected in the first instance and what they will tell you

b) identify what level of data you have for those variables

c) how you will analyse them and why; include any assumptions made or 'if x happens, we will do y'

d) consider the tests to achieve that

At the moment, there seems to be a somewhat random mix of approaches and analyses, with little justification.

Authors’ Responses:

Thank you very much for your comments and recommendations.

1. The run-on sentence has been deleted: “In order to ensure the greatest fairness among the different groups, a maximum of 15 and a minimum of 5 participants per group were established”.

2. The sample size has been calculated taking into account everyday cognition as the main variable, as you correctly stated. This was done by considering another similar study that also takes this variable as the main one. In both studies, a control group that undergoes traditional cognitive stimulation sessions is compared to an experimental group that undergoes sessions based on everyday cognition. The experimental group showed an improvement of 2.57 points compared to the control group's 0.39. In our study, we have chosen the PECC as the standardized test to measure the variable of everyday cognition.

We believe there may have been an error in the initial draft of the article we submitted. The data we will consider as the main dependent variables in the study are quantitative data derived from standardized tests. The only qualitative data, collected through an initial interview in the initial assessment of the participants, are the sociodemographic factors of the participants that will be analyzed as intervening variables. Therefore, we have removed the thematic analysis section as recommended and the following qualitative data section:

“Qualitative variables in the study will be defined by frequencies and percentages. A thematic analysis will be conducted to identify key patterns and themes within the qualitative data, such as the variable "occupation", collected through an interview”.

Regarding the raised question about the study variables, please find below the section of the article where the variables are explained: “The primary variable will be everyday cognition, measured by the Test for the Evaluation of Everyday Cognition (PECC) and cognitive performance, measured by the Montreal Cognitive Assessment Test (MoCA). Secondary variables will be functionality (Functional Independence Measure, FIM), emotional state (Yesavage Scale), frailty (Frailty Index) and independence in instrumental activities of daily living (Lawton and Brody Scale). We will also record the following intervening variables in the clinical history of each participant: social support (Questionnaire MOS) and adherence to treatment (Questionnaire ARMS-e), and other socio-demographic data (age, sex, level of education, marital status and main occupation)”.

It is true that we may have made an erroneous assumption that the distribution will be normal for some variables. 

Therefore, the following sentence has been removed:

“Sociodemographic variables and reference data, such as therapeutic compliance or level of social support, which have a normal distribution, will be defined by the mean and standard deviation, while variables which do not follow a normal distribution will be defined by the median and interquartile range”.

Being replaced by the following:

“Variables that have a normal distribution shall be defined by the mean and standard deviation, while variables that do not follow a normal distribution shall be defined by the median and the interquartile range”.

---

## [Editor Report · Decision Letter 2]

7 Mar 2024

Study protocol for a randomized controlled trial: Effect of an everyday cognition training program on cognitive function, emotional state, frailty and functioning in older adults without cognitive impairment

PONE-D-23-18288R2

Dear Dr. Sáez,

We’re pleased to inform you that your manuscript has been judged scientifically suitable for publication and will be formally accepted for publication once it meets all outstanding technical requirements.

Kind regards,

Francesco Sessa, Ph.D., MS

Academic Editor

PLOS ONE

Additional Editor Comments (optional):

Following the reviewers' comments, the authors improved their manuscript.
---

## [Editor Report · Acceptance letter]

22 Mar 2024

PONE-D-23-18288R2 

PLOS ONE

Dear Dr. Sáez-Gutiérrez, 

I'm pleased to inform you that your manuscript has been deemed suitable for publication in PLOS ONE. Congratulations! Your manuscript is now being handed over to our production team.

Kind regards, 

on behalf of

Lecturer Francesco Sessa 

Academic Editor

PLOS ONE